# Evaluation of Optic Disc, Retinal Vascular Structures, and Acircularity Index in Patients with Idiopathic Macular Telangiectasia Type 2

**DOI:** 10.3390/diagnostics13193046

**Published:** 2023-09-25

**Authors:** Muhammet Kazim Erol, Birumut Gedik, Yigit Caglar Bozdogan, Rojbin Ekinci, Mehmet Bulut, Berna Dogan, Elcin Suren, Melih Akidan

**Affiliations:** 1Department of Ophthalmology, Antalya Education and Research Hospital, University of Health Sciences, 07100 Antalya, Turkey; yigitcaglarbozdogan@gmail.com (Y.C.B.); rojbinekinci.1995@hotmail.com (R.E.); bulutme73@yahoo.com (M.B.); bernadoga3@hotmail.com (B.D.); elcin_baskan@yahoo.com (E.S.); 2Department of Ophthalmology, Serik State Hospital, 07500 Antalya, Turkey; birumut.gedik@gmail.com; 3Department of Ophthalmology, Kepez State Hospital, 07320 Antalya, Turkey; melcihhh@yahoo.com

**Keywords:** acircularity index, idiopathic macular telangiectasia type 2, optical coherence tomography angiography, optic disc vascular density, retinal vascular density

## Abstract

Background: We aimed to compare the retinal, optic disc vascular density (ODVD) values, and acircularity index (AI) of patients with idiopathic macular telangiectasia type 2 (IMT) and healthy individuals using the optical coherence tomography angiography (OCTA) device. Methods: The study included 39 patients with IMT and 37 healthy controls. The OCTA findings of the patients and controls were examined. Results: The total, parafoveal and perifoveal vascular density of the superficial capillary plexus, choriocapillaris blood flow, inside-disc ODVD, retinal nerve fiber layer (RNFL), and retinal thicknesses were found to be statistically significantly lower, and the foveal avascular zone value was statistically significantly higher in the IMT group compared to the control group (*p* = 0.001, *p* = 0.01, *p* = 0.02, *p* = 0.01, *p* = 0.009, *p* = 0.002, *p* = 0.02, respectively). There was a statistically significant negative correlation between best-corrected visual acuity (BCVA) and AI (*p* = 0.02), and a statistically significant positive correlation between peripapillary vascular density and BCVA (*p* = 0.04). Conclusions: We consider that the lower retinal, choriocapillaris, ODVD values, and retinal and RNFL thicknesses in the patients with IMT compared to the controls were due to vascular damage, remodeling, fibrosis, proliferation, and Müller cell damage. Ellipsoid zone defect, AI, and peripapillary vascular density are important indicators in the evaluation of visual acuity in these patients.

## 1. Introduction

Idiopathic macular telangiectasia type 2 (IMT) is a neurodegenerative disease characterized by bilateral, idiopathic, perifoveal retinal telangiectatic vessels and neurosensory retinal atrophy [1]. It was first described by Gass and Oyakawa in 1982 [2]. It usually occurs in the fifth and sixth decades of life. Symptoms include mild blurred vision, difficulty in reading, and metamorphopsia [3,4].

Biomicroscopic examination, optical coherence tomography (OCT), and fundus fluorescein angiography (FFA) are used in the diagnosis of IMT. On fundus examination, telangiectatic vessels starting from the parafoveal area, right-angle venules, intraretinal crystalline deposits, and photoreceptor atrophy are observed. The typical sign of IMT on FFA is hyperfluorescence in the parafoveal area, especially in the late phase. OCT shows intraretinal cystic hyporeflective spaces, hyperreflective plaques formed by the internal limiting membrane (ILM) and pigment clusters, foveal thinning, atrophy of the photoreceptor layer, and subretinal neovascularization [5,6,7]. Furthermore, other studies have detected retinal vascular blood flow changes in optical coherence tomography angiography (OCTA) in patients with IMT [8,9].

IMT is a vascular disease of unknown etiology. We consider that patients with this disease may present with changes in posterior segment vascular densities, and these results can show the vascular structure of IMT. When we reviewed the literature, we did not find any study that compared optic disc vascular density (ODVD) values between healthy individuals and patients with IMT. We aimed to compare the ODVD, retinal vascular density values, and acircularity index between patients with IMT and healthy individuals.

## 2. Materials and Methods

### 2.1. Patient Selection

The control group included 39 patients with IMT, who were followed up at the Ophthalmology Clinic of Health Sciences University Antalya Training and Research Hospital, and 37 healthy controls of similar age and gender, who presented to this clinic for a normal eye examination without any additional disease. The sample consisted of 39 eyes of the 39 patients and 37 eyes of the 37 healthy individuals, which did not have any of the following exclusion criteria: ocular pathology other than IMT (e.g., glaucoma, retinal vein occlusion, uveitis, epiretinal membrane diabetic retinopathy, and amblyopia), myopia higher than −6 diopters, history of eye surgery other than uncomplicated cataract surgery, and presence of systemic disease, which can contribute to retinal vascular changes (e.g., hypertension, diabetes mellitus, scleroderma, polyarteritis nodosa, Wegener’s granulomatosa).

Examination findings of the patients and controls were available in their files. The biomicroscopy and fundus examination findings, best-corrected visual acuity (BCVA) measurements, and FFA and OCT results of the patients and control groups were evaluated. In both groups, OCTA images acquired using the spectral-domain OCTA (AngioVue; Optovue, Inc., Fremont, CA, USA) device were also examined.

### 2.2. OCTA Evaluation 

The OCTA measurements of the patients were made at the 4.5 × 4.5 mm angio disc and 6 × 6 mm HD angio retina scales. Scans with a signal quality higher than 8/10 were evaluated. Images with a poor scanning quality due to motion and other artifacts were not included in our study. 

Foveal vascular density (FVD) was determined as the percentage of vessel density in a 1 mm circle centered on the fovea, total vascular density (TVD) in a 6 mm circle centered on the fovea, parafoveal vascular density (PAFVD) in a ring-shaped area between 1 mm and 3 mm, and perifoveal vascular density (PEFVD) in a ring-shaped area between 3 mm and 6 mm. These density measurements were obtained cross-sectionally using the automatic mode of the device. The ratio of the vascular image in these areas to the whole area provided the density value as a percentage.

The foveal avascular zone (FAZ) was calculated by the device in the automatic mode, and choriocapillaris blood flow (CBF) was recorded in mm^2^ based on blood flow in areas with a central radius of 1 mm and area of 3.142 mm^2^. The FAZ area (mm^2^), FAZ perimeter (mm), and acircularity index (AI) (the ratio between the measured perimeter and the perimeter of the same size circular area) were automatically obtained using AngioVue software (version Phase 7).

ODVD was measured from a 4.5 mm circle centered on the optic disc. Then, this density was determined as the percentage. Retinal nerve fiber layer (RNFL) thickness was automatically measured by OCTA from a 3.4 mm scanning circle centered on the optic disc. Retinal thicknesses between the retinal pigment epithelium (RPE) and İLM were automatically measured by OCTA. Choroidal thickness measurements were undertaken by two different observers using the “Enhanced HD line” section. The average of the measurements of the two observers was taken. The sclerachoroidal junction and RPE were taken as the boundaries for the subfoveal choroidal thickness (SCT) measurements.

OCTA grading was used in the evaluation of the extent of IMT. Accordingly, vascular anomalies in the temporal fovea in the superficial capillary plexus (SCP) and/or deep capillary plexus (DCP) were evaluated as OCTA grade 1, vascular anomalies in the nasal and temporal fovea in SCP and/or DCP as grade 2, diffuse vascular anomalies surrounding the fovea in SCP and/or DCP as grade 3, and neovascularization in the outer retina in addition to OCTA grade 1-3 findings as grade 4 [10] (Figure 1).

## 3. Statistical Analysis

All the data were analyzed using the Statistical Package for the Social Sciences, version 24.0 (SPSS Inc., Chicago, IL, USA). Descriptive statistics were given as standard deviation (SD), minimum and maximum values, mean, and median for numerical variables. The conformance of numerical variables to the normal distribution was evaluated with the Kolmogorov–Smirnov and Shapiro–Wilk tests. The independent samples *t*-test was used in the pairwise comparisons of data with a normal distribution, and the Mann–Whitney U test in the pairwise comparisons of those without a normal distribution. The Spearman correlation test was used in the correlations between the non-parametric numerical values. The results were evaluated at the 95% confidence interval, and *p* < 0.05 values were considered statistically significant.

## 4. Results

Thirty-nine patients with IMT and 37 healthy controls of similar age and gender participated in this study. In the IMT group, intraretinal hyporeflective cavitation was detected in 28 (71.7%) of the 39 eyes, ILM in 19 (48.7%), ellipsoid zone defect (EZD) in 24 (61.5%), and neovascularization in the outer retina layer in 3 (7.6%). The demographic characteristics of the two groups and the disease characteristics of the patients with IMT are given in Table 1.

Table 2 shows the DCP and SCP values in study groups. According to the results, the mean TVD, PAVFD, and PEVFD values of SCP were statistically significantly lower in the IMT group than in the control group (*p* = 0.01, *p* = 0.01, *p* = 0.02, respectively).

Table 3 shows the comparison of the FAZ, CBF, SCT, ODVD, and RNFL thickness values between study groups. No significant difference was found between the two groups in relation to the mean SCT, optic disc TVD, and peripapillary vascular density. However, the mean CBF, inside-disc vascular density of the optic disc, and RNFL thickness values were statistically significantly lower in the patients with IMT compared to the controls (*p* = 0.01, *p* = 0.009, *p* = 0.002, respectively). In addition, the mean FAZ value and acircularity index were statistically significantly higher in the IMT group than in the control group (*p* = 0.02, *p* = 0.01, respectively). 

Table 4 shows the comparison of retinal thicknesses between the study groups. The mean total retinal foveal thickness (TRFT), total retinal parafoveal thickness (TRPAT), total retinal perifoveal thickness, inner retinal parafoveal thickness, inner retinal perifoveal thickness, outer retinal foveal thickness, and outer retinal parafoveal thickness were found to be statistically significantly lower in the patients with IMT compared to the controls (*p* = 0.002, *p* < 0.0001, *p* = 0.006, *p* < 0.0001, *p* < 0.0001, *p* < 0.0001, *p* < 0.0001, respectively). Also, no significant difference was found between the two groups in relation to the mean inner retinal layer foveal thickness and outer retinal layer perifoveal thickness.

In the IMT group, the mean BCVA (±SD) was 0.4 ± 0.34 logarithm of the minimum angle of resolution (logMAR) (equivalent to the approximate Snellen BCVA of 20/50) (minimum: 1.3 (5/100), maximum: 0 (20/20)). The mean AI (±SD) of the IMT group was 1.14 ± 0.12 (minimum: 1.04, maximum: 1.56). Table 5 presents the correlations of BCVA and AI with OCTA measurements. Accordingly, while a statistically significant negative correlation was observed between BCVA and AI (*p* = 0.02), no correlation was found between BCVA and FAZ (*p* = 0.59). Peripapillary vascular density had a statistically significant positive relationship with BCVA and a statistically significant negative correlation with AI (*p* = 0.04 for both). There was also a statistically significant negative correlation between AI and DCP-PAFVD (*p* = 0.02). Figure 2 and Figure 3 present the graphs for the correlations between BCVA and AI and between BCVA and peripapillary vascular density, respectively.

In the IMT group, the logMAR values of BCVA were also examined according to the presence of intraretinal hyporeflective cavitation, EZD, ILM, membrane, and neovascularization in the outer retina layer and found to be statistically significantly higher in the patients with EZD compared to those without EZD (*p* < 0.0001) (Table 6).

## 5. Discussion

IMT is a vascular disease presenting with telangiectasia and aneurysmal dilatations in the vessels of the eye. Although its pathogenesis is not yet fully known, Müller cell degeneration and vascular remodeling are considered to play a role [1,3]. There is still ongoing research to elucidate the etiology of IMT.

Chidambara et al. compared 56 eyes of 28 patients with IMT and 10 eyes of 5 healthy individuals using OCTA and reported that the mean SCP-TVD and DCP-TVD values were 39.99 ± 3.99% and 39.03 ± 4.54%, respectively in the IMT group and 45.10 ± 0.84% and 44.21 ± 0.85%, respectively in the control group, indicating statistically significantly lower values in the former (*p* < 0.001 for both) [11]. Similarly, in our study, the SCP-TVD value was found to be statistically significantly lower in the patients with IMT compared to the controls.

Toto et al., comparing the OCTA measurements of the 15 eyes of 8 patients with IMT and 17 eyes of 17 healthy individuals, determined the mean SCP-FVD value as 24.74 ± 5.67% and the mean DCP-FVD value as 24.63 ± 5.89% in the IMT group, while the mean values of these measurements were 33.14 ± 9.99 and 34.21 ± 10.89%, respectively in the control group. The SCP-FVD and DCP-FVD values were statistically significantly lower in the patients with IMT compared to the controls (*p* = 0.005 for both) [10]. In another OCTA study, Park et al. compared 52 eyes of 26 patients with IMT and 40 eyes of 20 healthy individuals and reported the mean SCP-FVD and DCP-FVD values as 13.27 ± 4.47% and 22.58 ± 6.98%, respectively in the IMT group and 15.29 ± 3.53% and 32.25 ± 8.95%, respectively in the control group, revealing statistically significantly lower values in the patients with IMT (*p* = 0.027, *p* = 0.001, respectively) [8]. In the current study, although the SCP-FVD and DCP-FVD values were found to be lower in the IMT group compared to the control group, the differences were not statistically significant.

In their study, Toto et al. found the mean SCP-PAFVD value to be 47.06 ± 4.68% in the patients with IMT and 51.40 ± 3.33% in the control group and noted that this parameter was statistically significantly lower in the IMT group (*p* = 0.005). However, concerning DCP-PAFVD, no statistically significant difference was detected between the two groups [10]. Dogan et al., evaluating 40 eyes of 20 patients with IMT and 36 eyes of 18 healthy individuals using OCTA, reported the mean DCP-PAFVD value as 56.93 ± 2.27% in the former and 58.54 ± 2.20% in the latter. Thus, the DCP-PAFVD value was statistically lower in patients with İMT (*p* = 0.003) [12]. In another OCTA study, Ersoz et al. included 22 eyes of 22 patients with IMT and 24 eyes of 24 healthy individuals in the sample. The authors determined the mean SCP-PAFVD value to be 49.70% in the IMT group and 53.80% in the control group, indicating a statistically significantly lower value in the patients with IMT (*p* < 0.001) [9]. In our study, we found that the SCP-PAFVD and SCP-PEFVD values were statistically significantly lower in the IMT group compared to the control group, but there was no statistically significant difference between the two groups in relation to the DCP-PAFVD and DCP-PEFVD values. In addition, in our study, the inside-disc vascular density value of the optic disc and the RNFL thickness were statistically significantly lower in the patients with IMT compared to the controls.

Studies have suggested that the decrease in vascular density values in patients with IMT is due to the atrophy caused by increased intraretinal spaces in the vessels [11,13]. It has also been reported that vascular remodeling, proliferation, and fibrosis may be effective in this situation, and the death of Müller cells may play a role in the decrease in vascular density in patients with IMT [13,14]. Müller cells are responsible for intercellular connection and neuronal support in the retina. These cells are involved in retinal blood flow regulation, cytokine production, fluid-electrolyte exchange, and growth factor production. Due to these properties, Müller cells are effective in angiogenesis. Thus, damage to Müller cells may lead to photoreceptor atrophy, telangiectatic vessel formation, disruption of synapse formation in nerve conduction, and neovascularization [15,16,17,18].

Park et al. determined the mean FAZ value as 0.45 ± 0.12 mm^2^ in the IMT group and 0.27 ± 0.08 mm^2^ in the control group. The authors stated that the patients with IMT had a statistically significantly higher FAZ value than the controls (*p* < 0.001) [8]. Similarly, Dogan et al. reported the mean FAZ value to be statistically significantly higher in the IMT group (0.44 ± 0.13 mm^2^) compared to the control group (0.36 ± 0.09 mm^2^) (*p* < 0.009) [12]. Consistent with the literature, in our study, the FAZ value was statistically significantly higher in the patients with IMT compared to the controls. In addition, in our study, we determined the CBF value to be statistically significantly lower in the IMT group compared to the control group.

In a study by Toto et al., the IMT group was found to have a mean TRFT value of 214.13 ± 28.16 μm and a mean TRPAT value of 279.60 ± 15.67 μm, while these measurements were determined to be 258.18 ± 21.42 μm and 323.29 ± 13.33 μm, respectively in the control group, indicating statistically significantly lower values in the patients with IMT (*p* < 0.001 for both) [10]. In our study, the TRFT, TRPAT, total retinal perifoveal thickness, inner retinal parafoveal thickness, inner retinal perifoveal thickness, outer retinal foveal thickness, and outer retinal parafoveal thickness were all observed to be statistically significantly lower in the IMT group compared to the control group.

The decrease in retinal and choroidal blood flow due to vascular remodeling and vascular atrophy is considered to be effective in the increase in FAZ and the decrease in retinal thickness in patients with IMT. The development of IMT is accompanied by glial proliferation, with the formed spaces being filled by neuroglial cells [13,14]. In addition, it has been suggested that the fluid–electrolyte balance is impaired due to damaged Müller cells, and atrophy and apoptosis occur in photoreceptor cells [15,19,20]. Ersoz et al. showed a negative correlation between BCVA and AI and a positive correlation between the ellipsoid zone-RPE central foveal thickness and central foveal volume [9]. In another study, Park et al. detected a negative correlation between FAZ and BCVA [8], despite the absence of a correlation between these two parameters in the study of Ersoz et al. [9]. Spaide et al. observed temporal retraction and progressive displacement in macular vessels in the follow-up of patients with IMT [20]. Peto et al. found visual loss to be more common in IMT cases with multiple EZDs [15]. In our study, BCVA had a negative correlation with AI and a positive correlation with peripapillary vascular density. In the IMT group, the patients with EZD had a statistically significantly lower BCVA value than those without EZD. No correlation was observed between FAZ and BCVA.

Studies have shown recession in macular vessels, formation of vascular anomalies in the vessels in DCP and SCP, telangiectasia formation, and displacement of macular vessels in patients with IMT over time. It is considered that neuronal proliferation, foveal cavitation and migration, and consequent contraction may be effective in the displacement of the vessels. These mechanisms lead to the deterioration of visual acuity in patients with IMT. AI can be used as an indicator of these pathological changes in the vessels and has a correlation with visual acuity in this patient group [9,21,22]. It has been suggested that in patients with IMT, retinal blood flow regulation is disrupted, fluid–electrolyte balance is impaired, and photoreceptor cells undergo atrophy and apoptosis due to Müller cell damage. This leads to the development of EZD in these patients and negatively affects their visual acuity. Therefore, EZD can also be used as an indicator of visual acuity in patients with IMT [13,15]. In addition, we consider that due to vascular atrophy in these patients, the vascular structure that supplies blood to the optic disc is disrupted, and Müller cells are damaged. In our study, we found a positive correlation between peripapillary vascular density and BCVA, suggesting that the higher the peripapillary vascular density in patients with IMT; i.e., the lower the effect of vascular atrophy, the higher their visual acuity will be.

Concerning the limitations of the study, the number of patients was small. However, we consider that this study will guide future multicenter studies aiming to elucidate the vascular pathogenesis of IMT.

## 6. Conclusions

In conclusion, we compared the retinal, choroid, ODVD values, and acircularity index between the IMT and control groups. To our knowledge, this is the first study in the literature to compare ODVD values between healthy participants and patients with IMT. We consider that the lower retinal, choriocapillaris, RNFL, retinal thicknesses, and ODVD values in the IMT group compared to the control group are due to vascular damage, vascular remodeling, fibrosis, proliferation, and Müller cell damage observed in this disease. It can be suggested that AI, EZD, and peripapillary vascular density values are important indicators in the evaluation of visual acuity in patients with IMT.

## Figures and Tables

**Figure 1 diagnostics-13-03046-f001:**
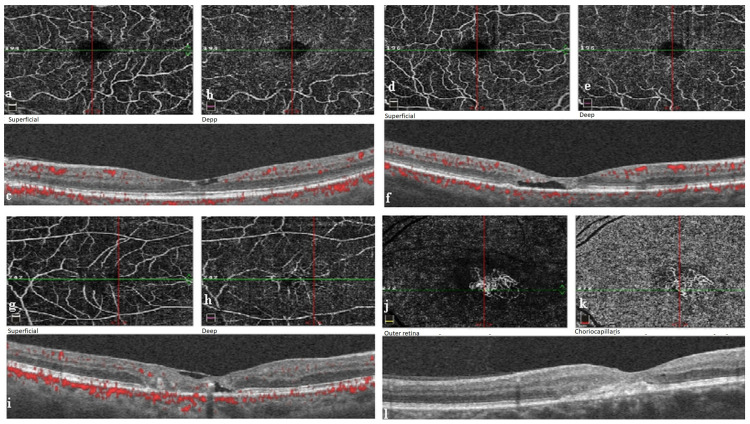
OCTA and OCT findings and OCTA grading in patients with idiopathic macular telangiectasia type 2: (**a**,**b**) spaces between vessels and telangiectatic vessels temporal to the fovea in the superficial and deep capillary plexus on OCTA (grade 1); (**c**) intraretinal hyporeflective cavitation and internal limiting membrane on B-scan OCT; (**d**,**e**) spaces between vessels and telangiectatic vessels in temporal and nasal fovea in superficial and deep capillary plexus on OCTA (grade 2); (**f**) intraretinal hyporeflective cavitation and ellipsoid zone defect on B-scan OCT; (**g**,**h**) spaces between the vessels surrounding the fovea and telangiectatic vessels in the superficial and deep capillary plexus on OCTA (grade 3); (**i**) intraretinal hyporeflective cavitation, internal limiting membrane, and ellipsoid zone defect on B-scan OCT; (**j**,**k**) neovascularization in the outer retinal layer on OCTA (grade 4); (**l**) neovascularization on B-scan OCT.

**Figure 2 diagnostics-13-03046-f002:**
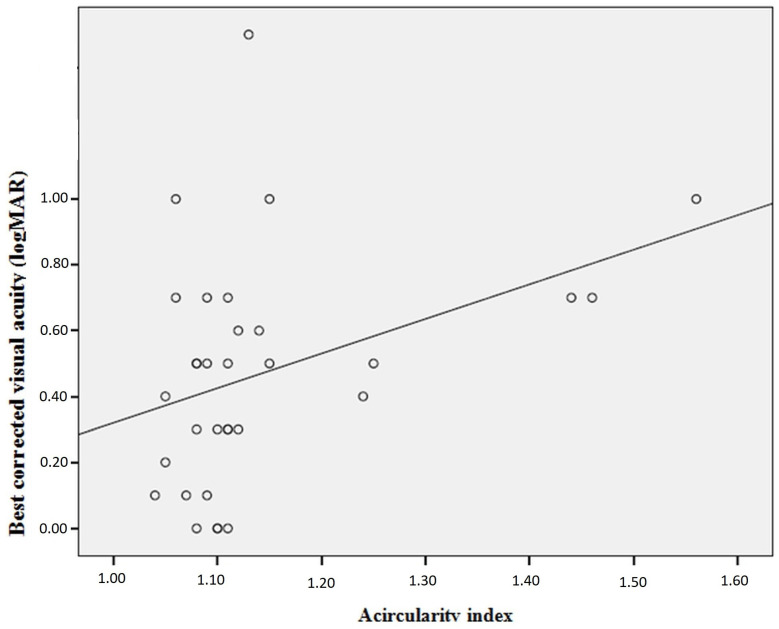
The correlation graph between best-corrected visual acuity and acircularity index.

**Figure 3 diagnostics-13-03046-f003:**
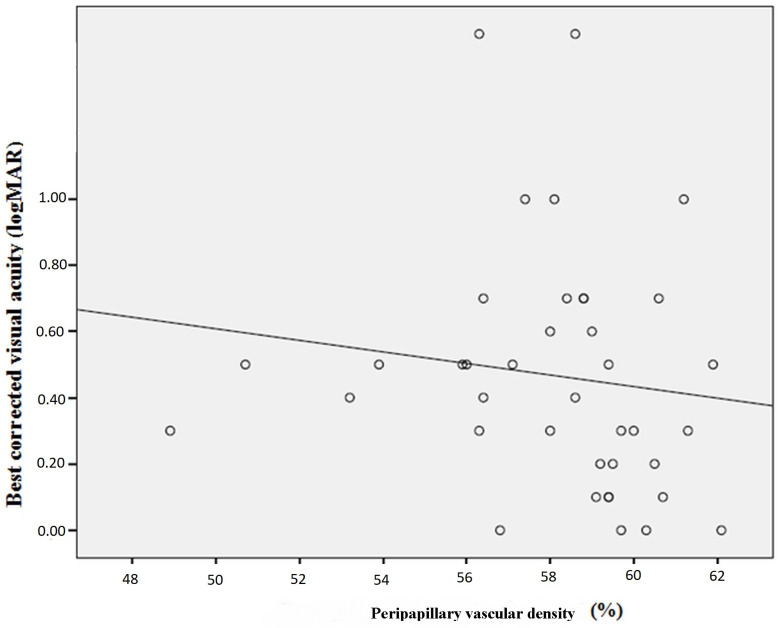
The correlation graph between best-corrected visual acuity and peripapillary vascular density.

**Table 1 diagnostics-13-03046-t001:** Demographic characteristics of the study groups and distribution of disease characteristics in the patients with idiopathic macular telangiectasia type 2.

Variable	IMT	Control
Mean age	63.36 ± 8.73	62.97 ± 8.94
Female/male	19/20	18/19
Right/left eye	20/19	19/18
Stage 1 IMT	12 (% 30.7)	
Stage 2 IMT	14 (% 35.8)	
Stage 3 IMT	10 (% 25.6)	
Stage 4 IMT	3 (% 7.6)	
Cavitation	28 (% 71.7)	
ILM	19 (% 48.7)	
Ellipsoid zone defect	24 (% 61.5)	
Neovascularization	3 (% 7.6)	
Total number of patients	39	37

IMT: idiopathic macular telangiectasia type 2. ILM: internal limiting membrane.

**Table 2 diagnostics-13-03046-t002:** Comparison of the superficial capillary plexus and deep capillary plexus vascular density values between the study groups.

Variable	Group	Mean	SD	Min	Max	*p*
SCP total vascular density (%)	IMT Control	46.93 49.77	0.57 0.47	36.8 41.9	53.1 56	**0.01 ****
SCP foveal vascular density (%)	IMT Control	17.21 20.21	1.24 1.11	4.2 5.8	40.2 32.7	0.06 **
SCP parafoveal vascular density (%)	IMT Control	47.29 51.31	0.95 0.64	26.9 38.5	56.7 57.9	**0.01 ***
SCP perifoveal vascular density (%)	IMT Control	47.86 49.86	0.58 0.54	38.6 39.7	56.1 57.5	**0.02 ****
DCP total vascular density (%)	IMT Control	48.88 49.28	0.87 0.87	39.2 35.7	59.5 62	0.86 **
DCP foveal vascular density (%)	IMT Control	35.07 36.16	1.68 1.19	15.6 20.4	56.2 51.1	0.52 **
DCP parafoveal vascular density (%)	IMT Control	54.92 53.13	0.80 0.63	44.60 44.60	64.3 61.8	0.06 **
DCP perifoveal vascular density (%)	IMT Control	49.62 50.62	0.96 0.95	38.3 34.3	60.6 64.4	0.56 **

Bold values represent statistical significance. IMT: idiopathic macular telangiectasia type 2. SD: standard deviation. Min: minimum. Max: maximum. SCP: superficial capillary plexus. DCP: deep capillary plexus. * Mann–Whitney U test. ** Student *t-*test.

**Table 3 diagnostics-13-03046-t003:** Comparison of the foveal avascular zone, choriocapillaris blood flow, subfoveal choroidal thickness, acircularity index, optic disc vascular density, and retinal nerve fiber layer between the study groups.

Variable	Group	Mean	SD	Min	Max	*p*
Foveal avascular zone (mm^2^)	IMT Control	0.32 0.25	0.01 0.01	0.15 0.10	0.55 0.41	**0.02 ***
Choriocapillaris blood flow (mm^2^)	IMT Control	1.95 2.08	0.01 0.01	1.69 1.92	2.24 2.27	**0.01 ****
Subfoveal choroid thickness (μm)	IMT Control	253.75 271.95	8.30 7.37	156 187	404 379	0.13 **
Acircularity index	IMT Control	1.14 1.08	0.12 0.02	1.04 1.03	1.56 1.13	**0.01 ***
Optic disc total vascular density (%)	IMT Control	55.55 55.78	0.41 0.37	45.80 49.50	58.70 59.80	0.98 *
Optic disc inside-disc vascular density (%)	IMT Control	57.79 59.97	0.57 0.57	50.50 51.30	66.90 67.40	**0.009 ****
Peripapillary vascular density (%)	IMT Control	58.11 58.32	0.43 0.41	48.90 50.10	62.10 63.40	0.92 *
Retinal nerve fiber layer (μm)	IMT Control	107.07 114.47	1.70 1.65	82 91	128 134	**0.002 ****

Bold values represent statistical significance. IMT: idiopathic macular telangiectasia type 2, SD: standard deviation, Min: minimum, Max: maximum. * Mann-Whitney U test, ** Student *t*-test.

**Table 4 diagnostics-13-03046-t004:** Comparison of retinal thicknesses between the study groups.

Variable	Group	Mean	SD	Min	Max	*p*
Total retinal foveal thickness (μm)	IMT Control	234.77 257.50	6.39 3.10	165 216	367 299	**0.002 ****
Total retinal parafoveal thickness (μm)	IMT Control	284.05 321.35	4.73 2.03	233 296	385 349	**<0.0001 ***
Total retinal perifoveal thickness (μm)	IMT Control	279.42 287.82	2.72 1.79	244 266	341 315	**0.006 ***
Inner retinal layer foveal thickness (μm)	IMT Control	67.70 72.02	2.82 1.67	32 53	119 97	0.17 **
Inner retinal layer parafoveal thickness (μm)	IMT Control	109.20 131.12	2.29 1.21	87 117	143 144	**<0.0001 ***
Inner retinal layer perifoveal thickness (μm)	IMT Control	107.10 113.35	1.19 0.95	92 101	126 126	**<0.0001 ****
Outer retinal layer foveal thickness (μm)	IMT Control	167.05 185.35	4.36 1.83	121 160	265 215	**<0.0001 ***
Outer retinal layer parafoveal thickness (μm)	IMT Control	174.87 190.35	2.84 1.38	145 173	254 211	**<0.0001 ***
Outer retinal layer perifoveal thickness (μm)	IMT Control	172.35 174.42	2.10 1.05	151 162	224 192	0.16 *

Bold value represents statistical significance. IMT: idiopathic macular telangiectasia type 2. SD: standard deviation. Min: minimum. Max: maximum. * Mann–Whitney U test. ** Student *t*-test.

**Table 5 diagnostics-13-03046-t005:** Correlations between the best-corrected visual acuity and acircularity index parameters and optical coherence tomography angiography measurements.

Variable	BCVA (logMAR)	*p*	AI	*p*
	R		R	
AI	0.402	0.02	-	
SCP total vascular density (%)	−0.061	0.71	−0.014	0.94
SCP foveal vascular density (%)	0.026	0.87	0.198	0.27
SCP parafoveal vascular density (%)	−0.079	0.63	−0.160	0.38
SCP perifoveal vascular density (%)	−0.087	0.59	0.028	0.88
DCP total vascular density (%)	−0.159	0.33	−0.304	0.09
DCP foveal vascular density (%)	0.135	0.41	0.322	0.07
DCP parafoveal vascular density (%)	−0.211	0.19	**−0.534**	**0.02**
DCP perifoveal vascular density (%)	−0.206	0.20	−0.288	0.20
Foveal avascular zone (mm^2^)	−0.089	0.59	−0.110	0.55
Choriocapillaris blood flow (mm^2^)	0.140	0.39	−0.33	0.85
Optic disc total vascular density (%)	−0.143	0.38	−0.275	0.12
Optic disc inside-disc vascular density (%)	0.28	0.86	0.288	0.10
Peripapillary vascular density (%)	**−0.318**	**0.04**	**−0.355**	**0.04**
Retinal nerve layer fiber thickness (μm)	−0.296	0.06	−0.147	0.42

Bold values represent statistical significance. BCVA: best-corrected visual acuity. logMAR: logarithmic minimum angle resolution. AI: acircularity index. SCP: superficial capillary plexus. DCP: deep capillary plexus. Spearman correlation test.

**Table 6 diagnostics-13-03046-t006:** Comparison of best-corrected visual acuity values according to the presence of intraretinal hyporeflective cavitation, ellipsoid zone defect, internal limiting membrane, and neovascularization in outer retinal layer in the idiopathic macular telangiectasia type 2 group.

Variable	Group	Mean	SD	Min	Max	*p*
Intraretinal hyporeflective cavitation	Present Absent	0.48 0.38	0.34 0.42	0.00 0.00	1.30 1.30	0.21
Ellipsoid zone defect	Present Absent	0.64 0.15	0.33 0.12	0.20 0.00	1.30 0.30	**<0.0001**
Internal limiting membrane	Present Absent	0.44 0.47	0.29 0.42	0.00 0.00	1.00 1.30	0.75
Neovascularization in outer retinal layer	Present Absent	086 0.42	0.55 0.33	0.50 0.00	1.30 1.30	0.08

Bold values represent statistical significance. IMT: idiopathic macular telangiectasia type 2. SD: standard deviation. Min: minimum. Max: maximum. Mann–Whitney U test.

## Data Availability

Data are contained within the article.

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
