# Peer review of "Evaluation of Optic Disc, Retinal Vascular Structures, and Acircularity Index in Patients with Idiopathic Macular Telangiectasia Type 2"

_diagnostics, 2023, doi:10.3390/diagnostics13193046_

Round 1

Reviewer 1 Report

1. The main question is; is there any differences in optic disc, retinal vacular structures in patients with idiopathic macular telangiectasia type 2 ? 2. The topic is original and relevant. There is no study on this issue in literature.  3. There is no other study on this issue. 4. No need any change in methodology. 5. Yes the conclusion is enough and consistent. 6. The references are apropriate. 7. No need any comment on the tabls and figures.

Author Response

Responses to Reviewers' Comments

The main question is; is there any differences in optic disc, retinal vacular structures in patients with idiopathic macular telangiectasia type 2 ? 2. The topic is original and relevant. There is no study on this issue in literature.  3. There is no other study on this issue. 4. No need any change in methodology. 5. Yes the conclusion is enough and consistent. 6. The references are apropriate. 7. No need any comment on the tabls and figures.

Response: We thank you for your comments.

Reviewer 2 Report

I read the paper entitled  “Evaluation of optic disc, retinal vascular structures and acircularity index in patients with idiopathic macular teleangiectasia type 2” very carefully and concluded that the paper is acceptable with minor revision in the present form for publication in your journal. The topic of the article is interesting. This study contributes to the retinal vascular density in patients with idiopathic macular teleangiectasia using OCTA using OCTA grading for the extent of idiopathic macular  teleangiectasia.

Some minor corrections must be made by the authors.

In Materiel and Methods page 2 row 66 at the end with the part of exclusion criteria …. and presence of systemic disease ... must be added: which can contribute to retinal vascular changes.

In Results on page 5, row162 must be added statistically not significant changes regarding table 4.

In Discussion part the reference must be corrected: the reference no 11 is citated pior the number 11.

Author Response

Responses to Reviewers' Comments

I read the paper entitled  “Evaluation of optic disc, retinal vascular structures and acircularity index in patients with idiopathic macular teleangiectasia type 2” very carefully and concluded that the paper is acceptable with minor revision in the present form for publication in your journal. The topic of the article is interesting. This study contributes to the retinal vascular density in patients with idiopathic macular teleangiectasia using OCTA using OCTA grading for the extent of idiopathic macular  teleangiectasia.

Some minor corrections must be made by the authors.

In Materiel and Methods page 2 row 66 at the end with the part of exclusion criteria …. and presence of systemic disease ... must be added: which can contribute to retinal vascular changes.

Response: We have revised this sentence as follows:

The sample consisted of 39 eyes of the 39 patients and 37 eyes of the 37 healthy individuals, which did not have any of the following exclusion criteria: ocular pathology other than IMT (e.g., glaucoma, retinal vein occlusion, uveitis, epiretinal membrane diabetic retinopathy, and amblyopia), myopia higher than -6 diopters, history of eye surgery other than uncomplicated cataract surgery, and presence of systemic disease which can contribute to retinal vascular changes (e.g., hypertension, diabetes mellitus, scleroderma, polyarteritis nodosa, Wegener’s granulomatosa).

In Results on page 5, row162 must be added statistically not significant changes regarding table 4.

Response: We have added this sentence to the manuscript:

Also, no significant difference was found between the two groups in relation to the mean inner retinal layer foveal thickness and outer retinal layer perifoveal thickness.

In Discussion part the reference must be corrected: the reference no 11 is citated pior the number 11.

Response: We have revised the references as follows:

 The first 9 references are in the introduction section. The reference no 10 is in the materials methods (OCTA evaluation) section and the next new reference number is number 11. The reference no 11 is in the discussion section.
